# Impact of Freezing and Freeze Drying on *Lactobacillus rhamnosus* GG Survival: Mechanisms of Cell Damage and the Role of Pre-Freezing Conditions and Cryoprotectants

**DOI:** 10.3390/foods14101817

**Published:** 2025-05-20

**Authors:** Junyan Wang, Peng Wu, Sushil Dhital, Aibing Yu, Xiao Dong Chen

**Affiliations:** 1Life Quality Engineering Interest Group, School of Chemical and Environmental Engineering, College of Chemistry, Chemical Engineering and Materials Science, Soochow University, Suzhou 215123, China; junyan.wang@monash.edu; 2Department of Chemical Engineering, Monash University, Clayton, VIC 3800, Australia; sushil.dhital@monash.edu (S.D.); aibing.yu@monash.edu (A.Y.)

**Keywords:** *Lactobacillus rhamnosus* GG, viability, survival, freezing, freeze drying, cell damage

## Abstract

*Lactobacillus rhamnosus* GG (LGG) is a common lactic acid bacteria used in the food industry with proven health benefits. Maintaining a high viability of probiotics during freeze drying and storage is crucial for their efficacy. The involvement of protectants and the optimization of operating conditions are promising techniques utilized to help bacteria microorganisms overcome environmental challenges. Although numerous studies have investigated the effectiveness of various protective agents in mitigating environmental stresses on bacterial cells and improving their survival during freeze drying, there is limited understanding of how freezing parameters impact the process by influencing ice crystal formation and bacterial cell microstructure. Therefore, this study systematically evaluates the effects of freeze-thawing and freeze-drying processes on the survival and metabolic activity of LGG. The results reveal that cell damage during freezing and freeze drying is a complex process influenced by a variety of physicochemical factors, including freezing conditions, sublimation and thawing processes, as well as the choice of cryoprotectants and reconstitution medium. Notably, freezing with water in liquid nitrogen at −196 °C resulted in the highest bacterial survival rate (90.94%) under short freezing durations, demonstrating the importance of freezing conditions. Freeze drying further reduced viability, with survival rates dropping to as low as 2% under suboptimal conditions. Interestingly, phosphate-buffered saline as a resuspension medium significantly increased the loss of viable LGG during both freezing and freeze drying. The addition of trehalose and skim milk as cryoprotectants enhanced survival to 15.17% post-freeze drying, emphasizing the role of protective agents in improving viability. This study provides novel insights into the critical role of freezing parameters and operational conditions in preserving probiotic viability, offering valuable guidelines for optimizing the freeze-drying process to maintain the functionality of probiotics.

## 1. Introduction

Probiotics, defined as live microorganisms that confer health benefits on the hosts when administered in adequate amounts [1], have been recognized for their role in intestinal microbiota modulation, gastrointestinal pathogen inhibition, immune system enhancement, and metabolic health improvement [2,3,4,5]. A high survival rate of probiotics during processing and storage is crucial in maintaining their health efficacy [6,7]. However, developing probiotic-containing food with adequate doses at the time of consumption is a challenge, as their viability might be affected during production, time spent in storage or shelf life [8].

Freeze drying is a conventional method used to preserve probiotic cells in a stable dried state at ambient temperature for prolonged periods [9]. It is a complex dehydration process comprising three stages: freezing, primary drying (sublimation) and secondary drying (desorption) [10,11]. Compared with high-temperature drying techniques, freeze drying is advantageous in avoiding general thermal damage to probiotic bacteria [12]. However, cell viability can still be compromised due to ice crystal formation, high osmolarity-induced membrane injury and macromolecule denaturation by dehydration [13]. Both biological factors (e.g., strain type, cell age, and initial cell concentration) and physicochemical parameters (e.g., cell growth conditions, freezing conditions, dehydration extent, and lyoprotectant selection) can influence the survival of bacterial cells during freeze drying [14,15,16]. The use of protective agents and the optimization of operational conditions have been identified as promising strategies to help bacteria overcome environmental challenges such as thermal, osmotic, mechanical, oxidative and acidic stress [17,18,19]. Stabilizing matrices typically incorporate sugars with low molecular weights and proteins with high molecular weights to form matrix microcapsule structures with probiotic bacteria [12].

While numerous studies have examined the effectiveness of various protective agents in mitigating environmental stresses on bacterial cells and improving their survival during freeze drying, limited attention has been given to how freezing conditions impact the process. Specifically, the influence of freezing on ice crystal morphology and bacterial cell microstructure remains understudied. The freezing rate, supercooling degree and ice nucleation temperature are three interdependent factors that have a direct impact on the freezing process [20]. Among them, the freezing rate is particularly crucial, as it directly influences the nucleation process, which has been reported as the most important step in controlling the size and distribution of ice crystals [20]. Thus, the pre-freezing temperature plays an important role in the freeze drying of probiotics, which must be controlled carefully. For example, Wang et al. (2020) investigated the impact of pre-freezing temperature on the survival of *Lactobacillus plantarum* strains and found that the optimal freezing temperature was strain-specific [13]. To the author’s knowledge, experimental results have only been obtained for limited bacterial strains, with *Lactobacillus rhamnosus* GG (LGG) notably absent from these studies. Therefore, an in-depth understanding of the effect of pre-freezing conditions on the growth of ice crystals and the microstructure of LGG bacterial cells during the freeze-drying process is urgently required.

The primary objectives of this study are to provide a mechanistic explanation for the inactivation of bacterial cells during freezing and freeze drying at the cellular level and to explore potential strategies to better retain their viability and activity. To this end, various operational parameters during the pre-freezing procedure, including different resuspension solutions, pre-freezing temperatures and durations, were systematically evaluated to better understand their impact on LGG survival during freeze drying. Furthermore, the effect of skim milk powder and trehalose as stabilizing matrices were investigated. This study is the first to systematically examine the influence of pre-freezing conditions specifically on LGG and offers new insights into how freezing parameters affect probiotic freeze drying. The findings will contribute to understanding the interplay between operational parameters and protective agents, offering insights to optimize the freeze-drying process for probiotics.

## 2. Materials and Method

### 2.1. Bacterial Strains, Medium and Growth Conditions

LGG was selected as a model bacterium to reveal the underlying mechanism of the freeze-drying process for LGG probiotics. Powdered LGG culture (ATCC 53103) was obtained from a commercial probiotic product (Culturelle^®^, CVS Pharmacy, New Haven, CT, USA). The LGG strain was retrieved by mixing one packet of Culturelle^®^ powdered culture with 10 mL of sterilized water. The resulting LGG suspension was then streaked onto a De Man–Rogosa–Sharpe (MRS) agar plate and incubated at 37 °C for 48 h. A seed broth was prepared by aseptically inoculating a flask of MRS broth with a single colony from the LGG culture medium, followed by incubation at 37 °C for 24 h. Subsequently, 1% (*v*/*v*) of the seed solution was transferred into another flask containing fresh MRS broth. The fermentation solution was obtained following 24 h of incubation at 37 °C.

### 2.2. Preparation of Cell Suspension and Feed Solutions

The fermentation solution was harvested and centrifuged at 5000× *g* at 20 °C for 10 min to remove the supernatant [21]. Feed solutions were prepared by thoroughly mixing the pellet with either water, PBS (0.01 M), or protective solution at a ratio of 1:1 (*v*/*v*). Skim milk powder (Devondale, Saputo Dairy Australia Pty Ltd., Victoria, Australia) and trehalose (BioDee, Beijing, China) were chosen as protective solutions and were rehydrated in distilled water at a concentration of 5% or 10% (*w*/*v*). Skim milk powder is recognized as an ideal material with a rigid structure forming a viscous layer to increase the viscosity of the solution and inhibit the growth of ice crystals during freezing [22]. Trehalose, on the other hand, serves as an effective protectant due to its unique ability to replace water molecules that are tightly bound to carbonyl groups [23]. The high viscosity and low mobility characteristics of trehalose can also help maintain the integrity of bacterial protein functions during the drying process [24]. The skim milk powder consisted of 35 g protein, 1 g fat, 53 g carbohydrate, 0.45 g sodium and 1.15 g calcium per 100 g solids. The feed solutions were then transferred into a stainless-steel container (inside diameter = 9.0 cm) for freezing. All mediums and solutions were prepared with Milli-Q water (Milli-Q^®^ Direct 16, Merck Millipore, Darmstadt, Germany). All chemicals used were of analytical grade.

### 2.3. Freeze Thawing and Freeze Drying

The prepared LGG suspensions were frozen using three techniques (i.e., refrigerator at −40 °C or −80 °C, controlled freezer at a cooling rate of 2 °C/min to −80 °C and liquid nitrogen at −196 °C), followed by either thawing or freeze drying. For uncontrolled-rate freezing, samples were stored at −40 °C or −80 °C in a freezer for 2 h, 24 h or 48 h. Controlled-rate freezing was performed at a cooling rate of 2 °C/min to −80 °C using a controlled freezer (LNC340, Tofflon, Shanghai, China). Low-temperature freezing was achieved by immersing LGG suspensions in liquid nitrogen (−196 °C) for 1 min followed by transferring to a freezer at −80 °C. The frozen samples were then either thawed in an incubator at 37 °C for 30 min or freeze dried in a freeze drier (FD-1C-80, BoYiKang Instrument Co., Ltd., Beijing, China) at −70 °C under a chamber pressure below 25 mbar. After 48 h of freeze drying, a manual crushing process was performed to convert the freeze-dried samples into powder for further analysis. Samples were carefully stored either at 4 °C or at ambient temperature. All experiments were independently performed in triplicate with the LGG samples freshly prepared for each experiment.

### 2.4. Determination of Survival Rate

The enumeration of LGG was carried out by the standard plate count method using 0.5% (*w*/*v*) peptone solution as the diluent. Either 1 mL of the LGG suspension or 1 g of the freeze-dried LGG powder was diluted in 10 mL of the peptone solution. After ten-fold serial dilutions, 0.1 mL of the appropriate dilution was spread on the MRS agar medium. Followed by a 48 h stationary incubation at 37 °C, microbial counts were measured as the colony forming unites (CFU) per plate. Moisture content (%) was determined by measuring the variation in dry mass (g) after drying in the oven at 105 °C until a constant weight was achieved. Viability tests were carried out in duplicate for each freeze-thawing or freeze-drying condition. Each measurement was carried out in triplicate and the average value was calculated.

The initial LGG viability in the feed suspension (*N*_0_, CFU/g) was calculated as(1)$$N_{0}=N_{f}/C_{f}$$
where *N*_*f*_ is the number of viable cells per milliliter (CFU/mL), and *C*_*f*_ is the concentration of the feed suspension (g/mL).

The viability of the freeze-dried LGG powder (*N*, CFU/g) was calculated as(2)$$N=\frac{N_{p}}{0.1\times \left(1-M_{w.p}\right)}$$
where *N*_*p*_ is the number of viable cells per volume of the reconstituted solution (CFU/mL), and *M*_*w*.*p*_ is the moisture content of the freeze-dried powder (%).

The survival rate of LGG (%) after freeze-drying was determined as(3)$$\%\;Survival\;rate=N/N_{0}\times 100$$

### 2.5. Measurement of Metabolic Activity of LGG

An iodonitrotetrazolium chloride (INT) assay, as described by Ulmer et al. (2000), was conducted with minor modifications to examine the metabolic activity of LGG after the freeze-thawing or freeze-drying cycle [25]. LGG suspensions were prepared by centrifuging at 5000× *g* for 10 min to remove the supernatant. A PBS solution (10 mM, pH 7.0) was prepared by dissolving 6.8 g H_2_KPO_4_, 0.1 g MgSO_4_·7H_2_O and 0.05 g MnSO_4_·H_2_O in 1000 mL of deionized water. The INT solution (4 mM) was prepared by dissolving the INT powder (Macklin Biochemical Co. Ltd., Shanghai, China) in the PBS solution. The LGG pellet was then washed twice with the PBS solution and suspended in 1 mL PBS. One mL INT solution was added to a final concentration of 2 mM. The mixture was incubated at 37 °C for 30 min to observe color changes. Metabolically active LGG cells could catalyze the reduction of colorless INT into red formazan dye, and the color intensity of the reaction mixture indicated the metabolic activity of each LGG sample [26]. In addition, the growth curve of LGG after freeze-thawing or freeze-drying treatment was analyzed to determine the change in growth capability. The fresh or thawed liquid LGG suspensions and the reconstituted solution of dried LGG powder were cultured in liquid MRS broth with an inoculum size of 1% (*v*/*v*). The optical density at 600 nm (OD_600_) was read every 30 min for 48 h using a microplate reader (M5, Molecular Devices, San Jose, CA, USA) at 37 °C. The measurements were carried out in duplicate and calculated to obtain an average value.

### 2.6. Scanning Electron Microscopy (SEM)

SEM was applied to study the microstructure of LGG cells. Double-sided carbon adhesive tape was used to place the freeze-dried powders on the SEM stub. After coating with gold, the samples were observed using equipment (Hitachi SU8230, Hitachi Ltd., Tokyo, Japan) operated at 15 kV with a working distance of 10 mm to visualize the morphology of the particles.

### 2.7. Statistical Analysis

All results were expressed as mean ± standard deviation (SD). Duplicates of each experimental condition were included to ensure statistical robustness. Statistical analysis was performed using SPSS version 3.0 (SPSS Inc., Chicago, IL, USA). The mean values were compared using analysis of variance (ANOVA) with a significance level of *p* < 0.05.

## 3. Results and Discussion

### 3.1. Impact of Resuspension Solvent

Although freeze drying is preferred in the preservation of probiotics, the long-term exposure of bacterial cells to extreme environments can still lead to a reduction in the viability of bacterial cells [13,27]. The cytoplasmic membrane of bacterial cells is recognized as the primary target of stress-induced injury. Its fluidity and integrity, as well as the fatty acid composition, are the key attributes affecting the survival of bacteria [27]. Because solvents strongly influence the physical and chemical properties of the cell membrane [28], the choice of solvents used in the resuspension of microbial pellets is a critical factor in the freeze-drying process of probiotics. It is assumed that solvents not only serve as a medium for the cells but can also provide osmotic balance and protection against stresses during freezing and sublimation.

In this study, deionized water and phosphate buffered saline (PBS, 0.01 M) were used to resuspend LGG pellets following centrifugation. Their impact on the survival of LGG during freeze thawing (Figure 1a) and freeze drying (Figure 1b) were investigated and compared. PBS is widely employed due to its buffering capacity and potential protective effect on cell membranes during processing [27]. However, as shown in Figure 1, the addition of PBS increased the loss of viable LGG during both the freeze-thawing and freeze-drying processes. During freeze thawing, 2 h of freezing led to a bacterial survival rate of 77.68 ± 3.29%, 77.19 ± 13.30% and 90.94 ± 12.82% for water at −40 °C, −80 °C and in liquid nitrogen, respectively. In contrast, PBS resulted in a lower survival of LGG compared to water, regardless of the freezing temperature. The most substantial reduction in survival was observed when freezing with PBS under −40 °C for 48 h, where survival decreased to only 6.60 ± 1.13%. In comparison, water showed less impact on LGG survival under prolonged freezing durations or insufficiently low freezing temperatures. Regarding the freeze-drying process, a significant reduction in the survival of LGG occurred regardless of the solvent used (*p* < 0.05) (Figure 1b). After 2 h of pre-freezing with water, approximately 12% of LGG survived freeze drying. However, when pre-frozen with PBS for the same duration, survival decreased sharply to approximately 2%. In addition, no LGG survived the freeze-drying process when pre-frozen with PBS at −40 °C for 48 h.

Different solvents can affect ice crystal formation by influencing water–water hydrogen bonds. In terms of PBS, the presence of salts can drive water from inside the cells to the extracellular environment through osmosis, promoting the formation of extracellular crystalline ice [29]. This removal of water from the cellular interior increases the solute concentration and consequently leads to high osmolarity-induced membrane damage. Therefore, PBS may not be an ideal choice as a resuspending medium during freeze drying. The extensive cell damage associated with PBS may arise from a combination of ice crystal formation, elevated intracellular solute concentration, and osmotic imbalance, which collectively compromise cell integrity and viability [30,31,32].

### 3.2. Impact of Freezing Temperature and Freezing Rate

Among the three stages of freeze drying, the initial freezing protocol is the most critical [33]. It has the potential to influence the survival of probiotic bacteria, given the fact that one of the major causes of cellular damage is the formation of intracellular ice [34]. Although freezing itself may not have lethal effect on cells, various physical stresses induced concurrently can damage them at low temperatures [14,34]. During freezing, the morphology and size distribution of ice crystals are fixed, which not only affects the subsequent drying behaviour including the heat and mass transfer during sublimation, but also determines the final properties of the dried product, such as physical state, residual moisture, reconstitution time and rehydration capacity [33]. The pre-freezing temperature and rate are important parameters that need to be controlled for enhanced survival during lyophilization [14]. Insufficiently low pre-freezing temperatures may result in incomplete freezing of the sample suspension, leading to expansion and foaming during the following sublimation process under vacuum conditions [13]. Conversely, when pre-freezing temperatures are excessively low, not only will the energy consumption be elevated, but also the survival of bacteria following freeze drying may be adversely affected [13]. Additionally, pre-freezing at an extremely slow rate facilitates the osmotic migration of intracellular water to the external environment, leading to the formation of extracellular crystals, the removal of extracellular water, increased solute concentration and thus osmosis imbalance, which collectively contribute to cellular stress and damage [35].

In this study, the LGG suspensions were frozen at different temperatures, including being exposed to liquid nitrogen (−196 °C), as well as temperatures of −80 °C and −40 °C after still standing for 30 min at 4 °C (Figure 2). Pre-freezing at 4 °C was employed to enhance resistance to lower temperatures and mitigate cold adaptation-induced viability loss [36,37]. Among the three freezing temperatures, liquid nitrogen consistently resulted in significantly higher bacterial survival rates, ranging from 71.31 ± 1.07% to 85.54 ± 16.44%, regardless of the freezing duration (*p* < 0.05). Freezing at −80 °C demonstrated a moderate performance during long-term storage (33.01 ± 20.91%–44.99 ±15.49%), whereas freezing at −40 °C resulted in survival rates of less than 7%. These findings align with prior studies indicating that lactic acid bacteria (LAB) exhibited better survival rates at lower freezing temperatures [38,39].

The freezing rate also played a pivotal role in LGG survival. Liquid nitrogen provided a higher cooling rate (196 °C/min) compared to the controlled freezing rate of 2 °C/min. As shown in Figure 2, bacterial survival was nearly doubled when the freezing rate increased from 2 °C/min to 196 °C/min, particularly for long-term freezing. This improvement can be attributed to the interaction between thermal flow and moisture outflow from the cells. A higher freezing rate could enhance heat and water flow rates, which may protect bacteria by preventing intracellular crystallization [15]. At higher freezing rates, the bacterial suspensions may immediately change to a microcrystalline state, in which physicochemical and biochemical reactions are suspended. In contrast, slower cooling rates promote the formation of larger ice crystals and osmotic water exit, resulting in greater cellular damage [15,36].

The relationship between bacterial survival and freezing temperature may depend on the type of microorganism. Extremely low pre-freezing temperatures or a high pre-freezing rate may also negatively impact microorganism viability. For example, ref. [14] investigated the impact of pre-freezing temperature on the survival of yeast and LAB isolated from wine and pointed out that freezing yeast in liquid nitrogen led to a lower survival rate compared to −20 or −80 °C. During freezing, rapid freezing prevents cells from effectively expelling water to maintain osmotic balance, resulting in the formation of intracellular ice crystals that can inflict significant structural damage or lead to cellular mortality [35]. Injury may occur due to the failure to migrate internal water to outside the cell and the formation of ice crystals inside the cell. The extent of this lethal damage is strain-specific [13,15], as different species exhibit variations in cell size, structure, and responses to freezing [40,41]. In general, microorganisms with smaller cell sizes and simpler cell structures tend to be more resistant to freezing damage. Their higher surface-to-volume (S/V) ratio facilitates faster water and heat transfer, reducing the likelihood of intracellular crystallization [15]. Bacteria, compared to yeast, have smaller cells and approximately five times the S/V ratio, providing them with better resilience against freezing-induced injuries. Additionally, differences in cell membrane composition between bacteria and yeast further contribute to their distinct freezing survival patterns [14]. Therefore, an appropriate pre-freezing rate should be selected before drying to ensure the formation of a glassy state rather than crystallization.

In addition, the difference between a controlled and uncontrolled freezing rate in the survival of LGG during freezing and freeze drying was examined. LGG suspensions were frozen either in a refrigerator at −80 °C with an uncontrolled freezing rate or cooled at a constant rate (2 °C/min) to −80 °C by gradient cooling. The changes in temperature under these two conditions were measured and recorded. As shown in Figure 3a, no significant differences were observed in the survival of LGG during freezing or freeze drying between the controlled and uncontrolled freezing rate (*p* > 0.05). The temperature profile of the freezer indicated a slower decrease in temperature during the first hour, followed by a faster rate in the second hour (Figure 3b). In contrast, the gradient cooling method maintained a steady decrease in the temperature throughout. These results suggest that while freezing rates influence survival under certain conditions, the specific impact on LGG may be minimal within the tested parameters.

### 3.3. Impact of Freezing Duration

LGG suspensions were frozen at different temperatures for 2, 24 or 48 h to explore the effect of freezing duration on the survival of bacteria after freeze thawing or freeze drying. The findings indicate that prolonged freezing durations negatively impact bacterial viability. For freeze thawing (Figure 2), the survival of LGG when pre-freezing for 2 h was significantly higher compared to a 24 h or 48 h duration (*p* < 0.05) regardless of the freezing temperature. Long-term freezing exerted a less negative impact on LGG pre-frozen with liquid nitrogen, with a reduction of only 14% when the freezing duration increased to 48 h from 2 h. At −80 °C, the bacterial survival was halved when the duration of freezing increased from 2 h (70.45 ± 4.85%–73.04 ± 4.96%) to 24 (33.01 ± 20.91%–36.7 ± 13.18%) or 48 h (37.45 ± 18.44%–44.99 ± 15.49%). The most pronounced reduction was observed at −40 °C, where survival rates plummeted from 62.30 ± 4.54% after 2 h to 6.51 ± 0.90% and 6.60 ± 1.13% after 24 and 48 h, respectively. Similar trends were observed in bacteria subjected to freeze drying (Figure 1b). Bacterial survival decreased with extended freezing times across all tested temperatures. However, the differences were not significant (*p* > 0.05). These results align with previous studies, which suggest that longer freezing durations exacerbate bacterial damage due to prolonged exposure to ice crystals and osmotic stresses [36,42].

### 3.4. Impact of Dehydration

In addition to freezing, dehydration during drying can also result in irreversible damage to bacterial cells. Water plays a critical role in maintaining the structure and functionality of the bacterial cell membrane. The removal of water during drying can lead to cell collapse and irreparable damage to the membrane [17]. A more pronounced reduction in the survival of LGG was observed after freeze drying (Figure 1b) compared to freeze thawing (Figure 1a), revealing that freeze drying had a more negative impact on bacterial cells than solely freezing. The finding is consistent with previous studies, indicating that most of the damage associated with freeze drying occurred during the drying procedure [13,14]. As an important component of the cell membrane, the phospholipid determines the integrity and behavior of the bacterial cell membrane [43]. Hydration assists bacterial cells in remaining intact by allowing interactions between water molecules and the polar group of phospholipid membranes [12]. However, the removal of water can cause the phase transition of phospholipids from a liquid crystalline phase to a gel phase, resulting in the leakage of the cell membrane [44].

In addition, changes in metabolic activity and growth capability of LGG bacteria during freeze drying were investigated through contact with INT and by measurements of OD_600_, respectively. The pink color in Figure 4a,b indicates the reduction in INT agent from metabolically active LGG cells. Freshly cultured LGG exhibited a more intense pink color, which is indicative of higher metabolic activity. In contrast, the metabolic activity of freeze-dried LGG was consistently lower compared to the untreated LGG regardless of pre-freezing conditions, as shown by the visibly lighter color observed after the reaction with INT. The relatively light color density demonstrated by the freeze-dried samples reflected that the freeze-drying process inhibited the generation of electrons to achieve the color change and reduced the metabolic activity of LGG [26]. Comparatively, LGG pre-frozen for 2 h (Figure 4a) showed a greater color intensity than that of LGG pre-frozen for 48 h (Figure 4b). This trend is in line with the survival data in Figure 1b, indicating that prolonged freezing negatively impacts bacterial metabolic activity. Interestingly, although the survival of LGG after the freeze-drying procedure was similar between 2 h and 48 h pre-freezing at −40 °C (11.42 ± 4.79% and 11.12 ± 2.01%, respectively), the difference in color demonstrates that the metabolic activity of LGG decreased with increasing pre-freezing duration. The discrepancy in the results between viable cell numbers and metabolic activity may be attributed to the viable but non-culturable state of LGG [45]. During freeze drying, both the mechanical force during water crystallization and the salt stress during solute concentration could lead to damages on the cellular envelope and cytoplasmic membrane [19]. However, the primary site of cellular injury and the extent of membrane compromise may be different depending on the treatment [46]. These results suggest that the metabolic enzyme systems of bacterial cells may still be disrupted during long-term freezing, even though there was no loss in viable cell count and the cell membrane remained intact [26,46].

In terms of optical density at 600 nm, it is a common approach to gauge the growth stage of bacteria, including a lag phase, an exponential growth phase and a stationary phase [46]. As shown in Figure 4c,d, freeze-dried LGG experienced a lag in entering the exponential growth phase compared to fresh liquid samples. This delay highlights the compromised growth capability of freeze-dried bacteria, consistent with the reduced metabolic activity observed in the INT assay. In conclusion, the freeze-drying process significantly impairs both the viability and functionality of LGG bacteria. While pre-freezing conditions play a role, prolonged freezing durations exacerbate the metabolic disruption, even when the cell viability appears unaffected. These findings underscore the importance of optimizing freezing and freeze-drying protocols to minimize damage to bacterial cells and preserve their metabolic and growth capabilities.

### 3.5. Impact of Cryoprotectants

To address the significant loss in viable LGG cells during freeze drying, trehalose and reconstituted skim milk (RSM) were used as cryoprotectants to increase the survival of LGG during a 24 h pre-freezing at −40 °C and 48 h freeze-drying process. Moreover, the morphology of freeze-dried LGG was visualized using SEM micrographs to theoretically investigate the inactivation mechanism of bacterial cells during the freezing and subsequent drying processes. Compared to a survival of approximately 11% for LGG after freeze drying with water and a survival of less than 2.5% with 0.01 M PBS, survival significantly increased to 15.17 ± 4.43% with the addition of 10% RSM and 5% trehalose (*p* < 0.05) (Figure 5). The addition of cryoprotective agents may improve the viability of LGG during freeze drying through two possible mechanisms. Firstly, cryoprotectants such as trehalose or skim milk form a glassy matrix with high viscosity and low mobility, which embed and immobilize probiotic cells for enhanced stability [47,48,49]. This matrix may inhibit diffusion, slow ice formation, and suspend deleterious reactions that compromise bacterial cell structures and compositions [50,51]. This was evident in the SEM micrographs (Figure 6), where LGG cells were trapped within the skim milk matrix. The cells exhibited visibly irregular shapes and surface cracks, indicating mechanical stresses endured during the freeze-drying process. Secondly, cryoprotectants exerted a water replacement effect. Sugars such as trehalose form stabilizing matrices that interact with cell membranes by replacing water molecules during dehydration. This interaction maintained the integrity of the membrane by forming hydrogen bonds between the hydroxyl groups of sugars and the phosphate groups on the phospholipid bilayer [12,21,52]. As previously discussed, dehydration can lead to the breakage of hydrogen bonds between phospholipids and the surrounding extracellular water. By interacting with the phospholipids through the hydrogen bond between hydroxyl groups of sugars and phosphate groups at the surface of the bilayer, cell death can be prevented by diminishing the negative effect on phospholipids and membrane integrity [52]. However, steric hindrance within membrane components and the inability of protectants to form sufficient hydrogen bonds can still lead to viability losses during dehydration [53].

In addition, the viability of freeze-dried LGG was further assessed over a 9-week storage period under various conditions. A significantly sharp decrease in viability was observed regardless of the type of protective agents used, where only approximately 0.0801 ± 0.0001% and 0.45 ± 0.13% of the freeze-dried LGG bacteria survived the 9-week storage duration at ambient temperature and 4 °C, respectively (*p* < 0.05) (Figure 7). The low viability of freeze-dried LGG after storage indicates that the addition of protectants such as skim milk and trehalose might not offer extra protection for the further survival of freeze-dried bacterial cells during storage. Moreover, there was no significant difference observed in the survival of LGG stored at different temperatures (*p* > 0.05), suggesting that low-temperature storage might not play a critical role in preserving the viability of freeze-dried LGG powder for extended durations. These findings emphasize the need for the further optimization of storage conditions and protective formulations to improve the stability of freeze-dried probiotics.

## 4. Conclusions

The present study has demonstrated that the mechanisms of probiotic cell damage during freezing and freeze drying are multifaceted and influenced by various physiochemical factors, including operational parameters during freezing, sublimation, thawing, and the type of cryoprotectants and reconstitution medium used. These factors play a crucial role in determining both the survival and metabolic activity of microorganisms throughout the freezing and freeze-drying processes. When undergoing the freezing and thawing cycle, LGG retained cell viability with survival rates ranging from 70% to 90% under most freezing conditions. However, the survival of LGG bacteria significantly decreased to below 13% during the freeze-drying process, together with a reduction in metabolic activity, revealing that most of the damage associated with freeze drying may occur during the drying phase. The choice of resuspension medium in the pre-freezing procedure had a considerable impact on the survival of bacterial cells after dehydration, as reflected by a significantly greater loss in viable LGG observed in PBS compared to water. The addition of trehalose and skim milk as cryoprotectants enhanced bacterial survival to 15.17% post-freeze drying. These findings underscore the importance of optimizing freezing and drying conditions to enhance bacterial survival. Further studies are required to optimize the operating conditions and improve bacterial survival during freeze drying. Strain-specific characteristics should be considered when determining the appropriate pre-freezing conditions before drying. While our study presents representative SEM images to illustrate morphological changes in freeze-dried LGG cells, a more detailed comparison of cellular structures before and after lyophilization, particularly under varying cryoprotectant conditions, would help elucidate the protective mechanisms of different cryoprotectants on probiotic cell morphology in future investigations.

## Figures and Tables

**Figure 1 foods-14-01817-f001:**
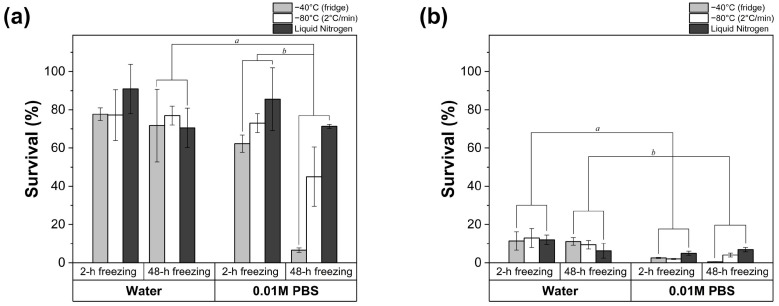
Survival of LGG after (**a**) freeze thawing or (**b**) 48 h freeze drying when pre-frozen for 2 h or 48 h using different freezing techniques (refrigerator at −40 °C, controlled freezer at a cooling rate of 2 °C/min to −80 °C, or liquid nitrogen at −196 °C) with different solvents. *a* and *b* represent statistical significance with *p* < 0.05.

**Figure 2 foods-14-01817-f002:**
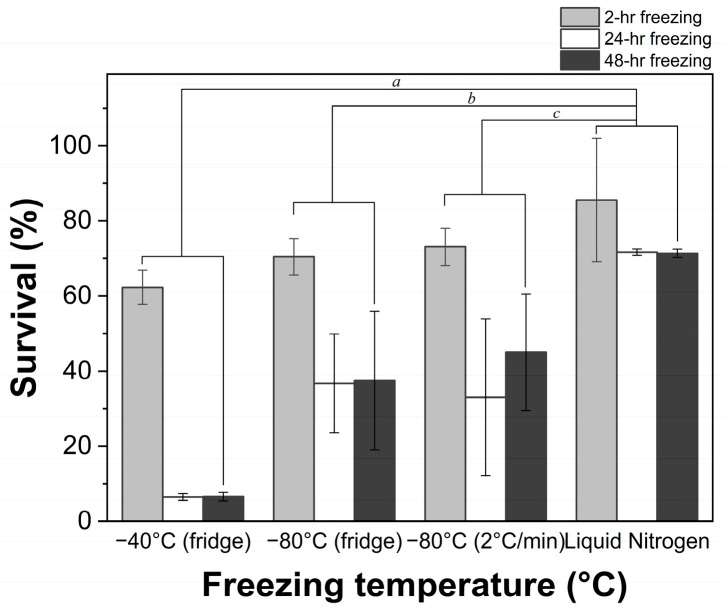
Survival of LGG after freeze thawing when pre-frozen for 2 h, 24 h or 48 h using different freezing techniques (refrigerator at −40 °C, refrigerator at −80 °C, controlled freezer at a cooling rate of 2 °C/min to −80 °C, or liquid nitrogen at −196 °C) with 0.01 M PBS. *a*, *b*, and *c* represent statistical significance with *p* < 0.05.

**Figure 3 foods-14-01817-f003:**
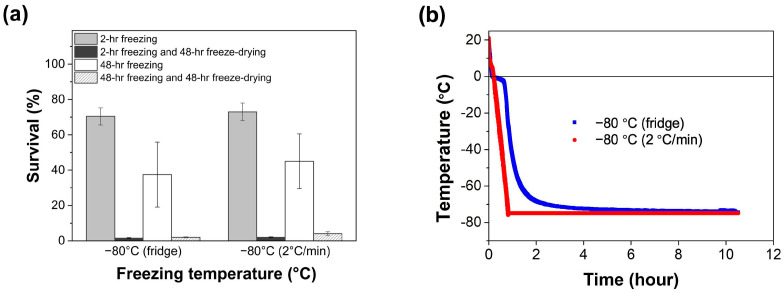
(**a**) Survival of LGG after 48 h freeze drying when pre-frozen by refrigerator at −80 °C or controlled freezer at a cooling rate of 2 °C/min to −80 °C for 2 h or 48 h with 0.01 M PBS. (**b**) Freezing rate of the refrigerator or controlled freezer to −80 °C.

**Figure 4 foods-14-01817-f004:**
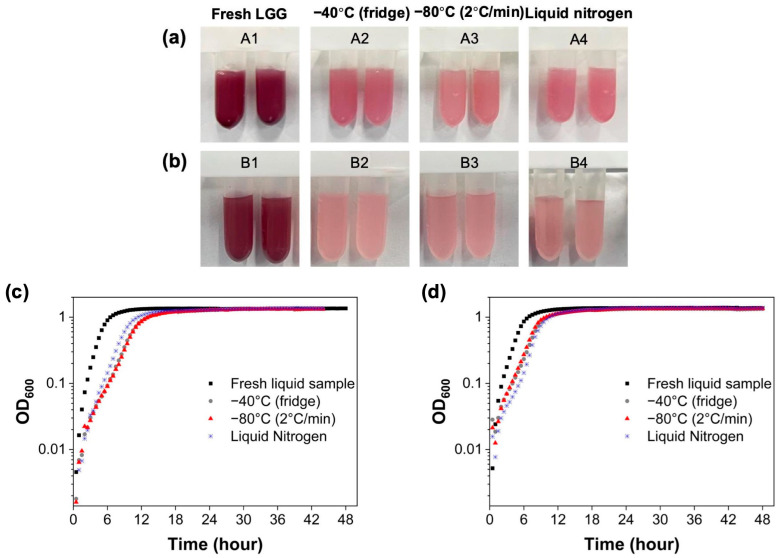
The activity of freeze-dried LGG pre-frozen using different freezing techniques after 48 h of freeze drying. (**a**,**b**) The metabolic activity of freeze-dried LGG pre-frozen for 2 h or 48 h, respectively, by INT (A1, B1: refrigerator at −40 °C, A2, B2: refrigerator at −80 °C, A3, B3: controlled freezer at a cooling rate of 2 °C/min to −80 °C, A4, B4: liquid nitrogen at −196 °C); (**c**,**d**) The growth curves of freeze-dried LGG pre-frozen for 2 h or 48 h, respectively, expressed as optical density at 600 nm by spectrophotometer.

**Figure 5 foods-14-01817-f005:**
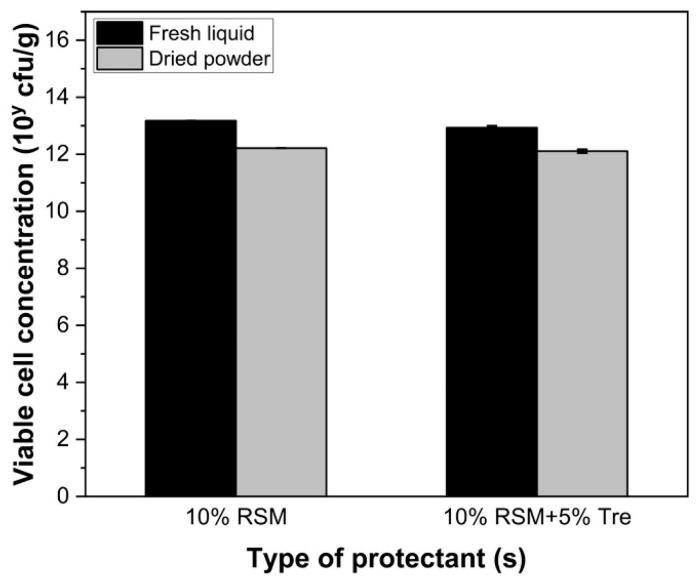
Changes in the viable cell concentration (log10 CFU/g) of LGG after 24 h pre-freezing at 40 °C and 48 h freeze drying with 5% trehalose and/or 10% RSM.

**Figure 6 foods-14-01817-f006:**
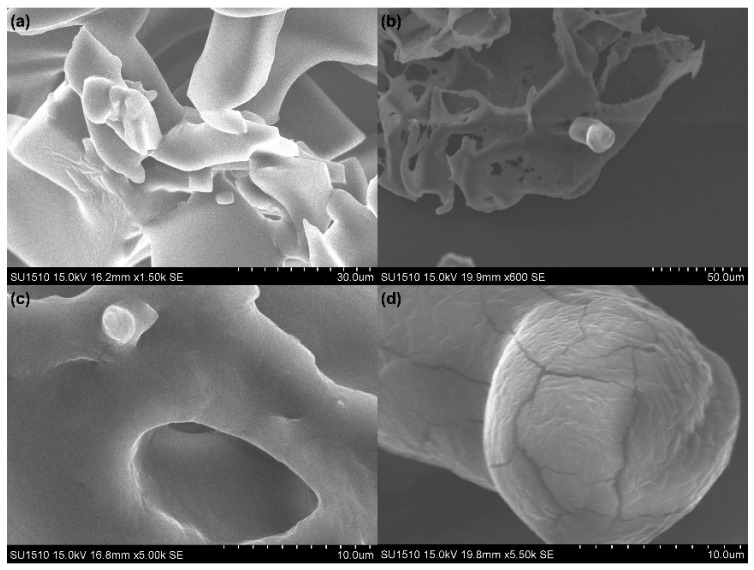
Morphology of LGG cell observed by SEM following 24 h freezing at −40 °C and 48 h freeze drying. (**a**) Freeze-dried LGG powder showed a cake-like porous structure. (**b**,**c**) LGG cells trapped within the skim milk matrix, partially protected by the cryoprotectant. (**d**) Structural damage and surface cracks observed on some LGG cells, suggesting membrane disruption during processing.

**Figure 7 foods-14-01817-f007:**
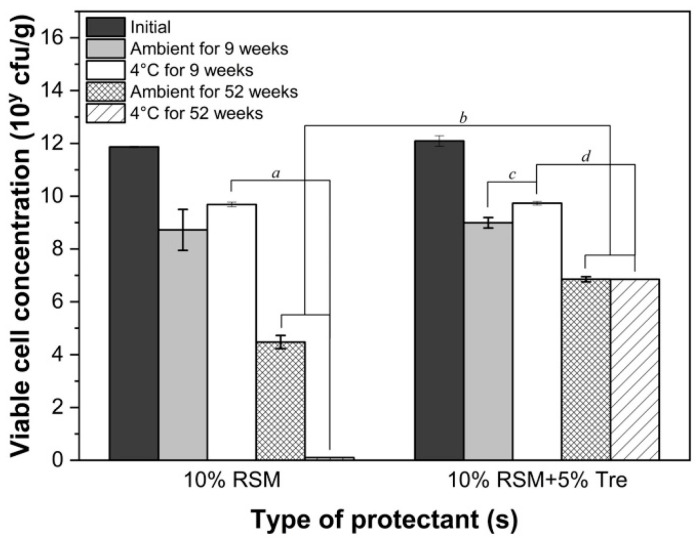
Changes in the viable cell concentration (log10 CFU/g) of freeze-dried LGG powder under different storage conditions. *a*, *b*, *c*, and *d* represent statistical significance with *p* < 0.05.

## Data Availability

The original contributions presented in this study are included in the article. Further inquiries can be directed to the corresponding authors.

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
