# Peer review of "Impact of Freezing and Freeze Drying on Lactobacillus rhamnosus GG Survival: Mechanisms of Cell Damage and the Role of Pre-Freezing Conditions and Cryoprotectants"

_foods, 2025, doi:10.3390/foods14101817_

Round 1

Reviewer 1 Report

Comments and Suggestions for Authors

In the manuscript by Wang et al. entitled "Impact of Freezing and Freeze-Drying on Lactobacillus rhamnosus GG Survival: Mechanisms of Cell Damage and Role of Pre-Freezing Conditions and Cryoprotectants", the authors present a study investigating the survival mechanisms of Lactobacillus rhamnosus GG (LGG) during freezing and freeze-drying, with a focus on the impact of pre-freezing conditions and cryoprotectants. LGG, a probiotic with health benefits, faces viability challenges during these processes, which was also an issue that stimulated this study. The study systematically investigates how freezing parameters such as temperature and duration of freezing affect the survival and metabolic activity of LGG. The results show that freezing with liquid nitrogen at -196°C for a short duration gives the highest survival rate (90.94%), while freeze-drying significantly reduces viability, with survival rates dropping to as low as 2% under sub-optimal conditions. Phosphate-buffered saline as a resuspension medium increases cell loss, while trehalose and skimmed milk as cryoprotectants increase the survival rate to 15.17% after freeze-drying. The study provides insights into the optimisation of freezing and drying conditions to preserve the functionality of probiotics and highlights the importance of operational parameters and protective agents in maintaining the viability of LGG during freeze-drying.

Although the topic covered is quite narrow, the paper may be of interest to a certain number of readers of the journal. Otherwise, the topic is contemporary and the work can be considered novel. On the technical side, the general results seem reliable, considering that standard experimental techniques are used in the study. The paper itself is generally well written and readable, although there are two places in the manuscript where the text could be improved and clarified (see below).

In view of the above, I recommend publication of the manuscript in Foods journal after a minor revision, in which the authors should take into account my comments below:

1.) The work “(Division, 2006)” mentioned in line 41 cannot be found under References.

2.) The reference “Polo et al., 2017” is mentioned twice in the same parenthesis [line 61].

3.) Change the term “amount of viable cells” [line 160] to “number of viable cells”. The latter term was already used in line 156 to describe the similar term Nf (which has the same units as Np). The problem also lies in the fact that in chemistry the term “amount" has a different meaning than “number”.

4.) If the sentence “The measurements of absorbance were carried out in duplicate and calculated to obtain an average value.” [lines 182-183] is a continuation of the description in the preceding sentence “The optical density at 600 nm (OD600) was measured every 30 min for 48 h using a microplate reader (M5, Molecular Devices, USA) at 37°C.” [lines 180-182] and the term “absorbance” is to be used (as a synonym) for the term “optical density“, then the use of the term “absorbance” is inappropriate (the terms “absorbance” and “optical density“ are not synonyms). It could be that the term “absorbance” is used on the M5 microplate reader instrument, but what was measured in this particular case is not absorbance. In fact, the main reason for the weaker intensity of light reaching the detector of the instrument is not the absorption of light in suspension, but the scattering of light. While the term optical density (OD) is a broader term that can include not only the absorption of light in a sample but also other factors such as scattering (which should be the main reason for the decrease in the intensity of light reaching the detector) and reflection, the term absorbance refers only to the absorption of light.

If the sentence “The measurements of absorbance were carried out in duplicate and calculated to obtain an average value.” [lines 182-183] refers to something other than the measurements described in lines 180-182, then the absorbance measurements must be described in more detail (samples, instrument, wavelength).

5.) The viability given in line 442 (“0.08 ± 0.000079%”) cannot be determined so precisely. The error must be much higher than "± 0.000079%”.

6.) In MDPI journals, the referenced works are given in the text with numbers (https://mdpi-res.com/data/mdpi_references_guide_v9.pdf) and not with the surnames of the authors and the years of publication of the works.

7.) Please also note the instructions for authors on how the references should be described in References.

Author Response

Dear Reviewers,

We wish to thank the reviewers for taking the time to review our manuscript (foods-3617137) and for providing us with informative comments and constructive suggestions. In this response, we have thoroughly reviewed the manuscript and carefully considered each point raised by the reviewers, incorporating your valuable suggestions into the revised version of the manuscript. To improve the clarity and robustness of the study, we have made the necessary revisions, which are highlighted in red in the revised manuscript for ease of review. We hope that the extensive revisions have met the standards required for publication in the journal. However, it would be our great pleasure to make any further revisions if necessary to ensure that this study achieves its maximum impact.

Please find the detailed responses to each comment as attached.

Kind regards,

Peng WU

Reviewer 2 Report

Comments and Suggestions for Authors

The manuscript titled "Impact of Freezing and Freeze-Drying on Lactobacillus rhamnosus GG Survival: Mechanisms of Cell Damage and Role of Pre-Freezing Conditions and Cryoprotectants" addresses an important gap regarding the impact of pre-freezing conditions on probiotic viability, which is essential for the production and long-term stability of probiotic-containing foods. Below are some comments and recommendations for improving the manuscript

  1. In this study, INT assay and OD600 measurement are used to evaluate the metabolic activity after lyophilization. However, the observed discrepancy between viable cell numbers and metabolic activity has not been analyzed in detail. Consider discussing possible causes such as enzyme inactivation or membrane damage.
  2. In Figure 6, detailed SEM images showing the morphology of LGG cells before and after lyophilization with and without cryoprotectants are needed. This visual comparison will support the cellular integrity argument.
  3. Correct typographical issues such as “metabolic health improvment” → “improvement”.
  4. Consider merging Figures 4 and 5 to present the metabolic activity and viability trends more cohesively.
Comments on the Quality of English Language

The English could be improved

Author Response

(The authors gave the same response as above.)
